# Parent–Child Relationships, Digital Media Use and Parents’ Well-Being during COVID-19 Home Confinement: The Role of Family Resilience

**DOI:** 10.3390/ijerph192315687

**Published:** 2022-11-25

**Authors:** Marina Everri, Mattia Messena, Finiki Nearchou, Laura Fruggeri

**Affiliations:** 1School of Medicine, University College Dublin, D04 V1W8 Dublin, Ireland; 2School of Psychology, University College Dublin, D04 V1W8 Dublin, Ireland; 3Centro Bolognese di Terapia della Famiglia, 40121 Bologna, Italy

**Keywords:** parents’ well-being, technology interference, family resilience, parental stress, marital conflict, digital media, ICT, COVID-19, lockdown

## Abstract

Research has provided substantial evidence on the role of parents’ well-being in the quality of parent–child relationships and children’s adjustment. Parents’ stress and parental couple conflict have been linked to children’s adverse developmental outcomes. However, little is known about the factors that affect parents’ well-being when coping with multiple stressors such as those brought by the recent COVID-19 global pandemic. Our study intended to examine the predictors of parental well-being by looking at the contextual factors of COVID-19 home confinement, i.e., the use of digital media and parents’ domestic workload, and family resilience in two countries: Ireland and Italy. Additionally, the age and number of children were controlled as potential variables impacting parents’ well-being. A three-step hierarchical regression analysis was applied. The results showed that family resilience was a very strong predictor of parents’ well-being after controlling for any other variable. Parental couples’ conflict over the use of technology predicted lower levels of parents’ well-being, while, notably, parent child-conflict and domestic workload were not associated with parents’ well-being. Additionally, the age of children did play a role: the higher the mean age of children in the family the better the parents’ well-being. The findings are discussed in the light of cross-country differences and their implications for research and practice.

## 1. Introduction

Substantial literature on parent–child relationship has addressed the implications of parents’ conflict, stress, and well-being for children’s social and emotional adjustment across the life-course [1,2,3]. For instance, the transition to parenthood can be a highly stressful event for couples: pre-transition conflicts between partners can be amplified and affect co-parenting tasks with consequences for children’s well-being [4,5,6,7,8]. Similarly, adolescence can be a period of stress and conflict for parents and children. They are, in fact, called to renegotiate families’ rules and roles, power, and interpersonal distances [9,10,11]. Additionally, as indicated by ecological [12] and systemic models [13], historical events, together with the characteristics of social-cultural context (e.g., economic hardship, [14,15]), can have an impact on parent–child relationships as well as on parents’ and children’s well-being. The recent COVID-19 global pandemic can be considered as a historical catastrophic event that has affected individuals’ well-being and their significant relational contexts. In fact, research carried out during the pandemic has extensively examined parental stress and associated risks, such as parental burnout [16], and the implications for parent–child relationships as well as for children’s and adults’ mental health, especially during home confinement and isolation [17,18,19,20]. Families with young children experienced more challenges compared with adults with older or no children due to the closure of schools and childcare services [21,22]. This resulted in increased pressure on parenting and additional negotiations between partners for managing childcare and domestic workload [23]. Parents’ and children’s home confinement has also implied an increased use of digital technologies for maintaining social connections and communication during isolation [24,25]. This has afforded continuity with contexts outside the family household such as schools, healthcare, and emergency services. However, the necessary continual connection to Internet and the use of digital devices (computer, tablets, smartphones) for school and work activities have exacerbated conflict among family members [26]. In this scenario, family resilience [27], namely the capacity of families to heal and adapt or even grow stronger when devastated by traumatic events, was found to be a protective factor for parents’ well-being [28]. However, while studies on resilience, family conflict, parental stress, and well-being during the different waves of the COVID-19 pandemic have yielded important evidence, there is limited research that has examined the role of digital technologies as a key contextual factor affecting parent–child relationships, parents’ wellbeing, and ultimately children’s and adolescents’ adjustment.

### 1.1. Parent–Child Relationships and Parents’ Well-Being during COVID-19 Home Confinement

Studies on parent–child relationship carried out during the COVID-19 pandemic have given particular attention to parental stress and well-being-related dimensions, e.g., [29,30,31,32]. Parents’ individual well-being, as well as the quality of the parental couple relationship, emerge as important factors in defining the quality of parent–child relationship and children’s adjustment [5,10,33,34]. Katz’s and Gottman’s studies showed that marital conflict has a spillover effect on parenting and on children’ emotional regulation and peer interactions [1,2]. Similarly, Cui et al. [3] found an association between marital problems and adolescents’ internalising and externalizing behaviours. Other studies considered contextual variables affecting parents’ well-being, especially under stressful circumstances. McLoyd [15] showed that economic hardship impact parents’ mental health and self-efficacy, which in turn affect their parenting and the quality of care provided to their children [3]. Consistently, more recent studies on COVID-19 pandemic have shown that adults with children had greater levels of stress compared to adults without children [19,35,36]. Motherhood, psychological distress, and having younger children predicted higher parenting-related exhaustion [37]. Parents with preschool children perceived more stress related to staying at home during the quarantine and doing repetitive activities every day [22]. In families with adolescents, the impact of home confinement had consequences for children’s mental health in terms of increased anxiety and depression; however, the increase of these symptoms was correlated with parental stress [20]. Family structure variables, such as the number of children living at home, have also being associated to higher level of parental stress during home confinement [38]. Furthermore, parents in families already dealing with multiple stressors were at greater risk than others [39,40]. Not surprisingly, research showed that domestic violence between parents and between parents and children increased during quarantine [41,42]. Similarly, dysfunctional couple relationships were amplified by lockdown restrictions and home confinement [43].

Workload and childcare have been identified as other key variables impacting the quality of parent–child relationship and parental couple’s well-being. For instance, Seiz [23] found that the pandemic allowed couples to renegotiate the traditional division of labors. Similarly, Yerkes et al. [44] showed that gender inequality in the division of childcare and household work decreased during lockdown; however, disagreements between partners regarding the division of childcare tasks increased among parents with children in primary schools compared with parents with children in secondary schools. Taken together, these studies show that the unprecedented circumstances of COVID-19 pandemic have both increased parental stress and amplified dysfunctional family dynamics, with detrimental consequences for parents’ and children’s relationships and well-being.

### 1.2. COVID-19 Contextual Factors: Digital Media and Family Conflict during Home Confinement

The use of digital technologies is now embedded in families’ everyday life tasks and routines [45] and parent–child communication [46,47,48]. COVID-19 lockdown has exacerbated the dependency on digital devices for communication exchanges, as well as for tasks and routines that would have regularly happened face-to-face: work and school activities, connection with extended family and friends, shopping, celebrations, etc. [25]. Therefore, it was not surprising that the amount of time that family members spent using digital technologies had increased during lockdown [49]. This phenomenon has offered parents and children new opportunities to use online platforms and online services together for healthcare, work, and education. However, this has increased screen time, thereby also implying possible exposure to correlated risks and harm [50,51]. Among the risks, researchers found that *technoference* or technology interference, namely the interruption of face-to-face communication due to technology [52,53], characterized family interactions during home confinement and isolation with negative consequences for children’ social competence [26]. In general, *technoference* was associated with poorer quality of partner relationship, co-parenting, and children’s behavior problems [52,54]. Consequently, home isolation, as well as parents’ and children’s increased use of digital media, may have increased *technoference*, which in turn may have amplified family conflicts. However, research has not explored how these contextual variables of the pandemic—the substantial use of digital media and the interference of devices in face-to-face communication—might have impacted on parents’ well-being.

### 1.3. COVID-19 and Family Resilience

Family resilience has been considered central in buffering against COVID-19 risks and the multiple losses caused by the pandemic [27,55,56]. The conceptual framework of family resilience [55] builds upon the systemic principle that a traumatic event has an impact on individuals and reverberates through families’ mutual relationships. Therefore, the capacity of families to overcome crisis and rebound from traumatic events, such as a global pandemic, is not only beneficial for individuals’ health, but it also triggers relational processes that happen among family members. Walsh [27] identified key family processes concerning three dimensions of family functioning (shared belief systems, organizational resources, and communication processes) and three subdomains for each dimension. In other words, the ways in which families face trauma, loss, and varied situations of adversity depend on their values and belief system, on the ways in which family members communicate and share emotional support, and on the capacity to mobilize internal and external resources. Prime, Wade, and Browne [57] emphasized the centrality of family resilience dimensions of shared belief systems and communication and emotional support to promote family resilience during the pandemic as a way to protect caregivers’ well-being and ultimately children’s adjustment. Similarly, Walsh [56] proposed considering a shared belief system as a key dimension for positively coping with the multiple losses caused by COVID-19 pandemic. However, Coulombe et al. [58] have found that the buffering effect of resilience was limited for the multiple stressors associated with COVID-19. In fact, the literature on the ‘healing potential of family resilience’ is predominantly characterized by conceptual works that have guided clinical interventions and that have rarely assessed the perceptions of parents and children. To address this gap, our study investigated family members’ perceptions of family resilience dimensions while families were experiencing restrictions measures.

### 1.4. Aims of the Study

The main aim of this study was to examine the well-being of parents having at least one minor child (<18 years of age) and living in the Republic of Ireland and Italy during the first COVID-19 lockdown. More specifically, we aimed to investigate (a) the impact of digital media use and conflict over the use of digital devices (*technoference*) on parents’ well-being, and (b) the role of family resilience, a key protective factor for coping with adversities and fostering family members’ well-being. Additionally, existing evidence showed that given the circumstances of COVID-19 restrictions, domestic workload and childcare had an impact on parents’ stress and well-being-related dimensions e.g., [29,30,31,32]. Furthermore, having young children [37] and the number of children in the household contributed to higher parental stress and reduced well-being during the first phase of the pandemic [38]. Therefore, (c) we also examined the impact of children’s age, and the number of children present in the household on parents’ well-being.

## 2. Materials and Methods

### 2.1. Context and Procedures for Data Collection

Data for this study were collected during the first lockdown of the COVID-19 pandemic when both participating countries, Italy and the Republic of Ireland (These countries participated in a larger research project on family resilience that emerged from the collaboration between two research teams based in Ireland, University College Dublin and Italy, Centro Bolognese di Terapia della Famiglia) were under high level restrictions (April–June 2020). Italy was one of the first countries to be affected by the wide spread of COVID-19 after China, which was in early February 2020. In the Republic of Ireland, by March 2020 restrictions were in place, including stay-at-home orders (except for essential workers, shopping, medicines, exercise, and care for relatives) [59]. Lockdown measures in both countries included bans on public and private gatherings, and closures of non-essential shops, community centers, bars, and restaurants. In Italy restrictions started to be eased on 4 May 2020, whereas in the Republic of Ireland they began easing on 18 May 2020. Despite being lifted earlier, restrictions in Italy were stricter compared with restrictions in Ireland: in some Italian regions people were not allowed to a distance from their house for more than 200 m. In Ireland, the walking distance from one’s house was 2000 m. After May 2020, significant limitations on non-essential travel, public events, and schools and universities activities remained in place in both countries.

For the purposes of the present study, parents with at least one minor child (<18 years of age) were recruited through schools and educational services in different regions in Northern Italy, and through an online platform in the Republic of Ireland. Anonymous questionnaires (The questionnaire was developed first in Italian and then translated in English. Three authors are native Italian speakers but have worked and lived in English-speaking countries for more than 5 years. A first level translation was carried out by the authors, then a native English-speaker colleague was involved for refining the translation in English) were distributed and administered online through the survey platform Qualtrics. Informed consent from participants was obtained, asking them to fill out an online consent form before starting the questionnaire. All questions included in the questionnaire referred to the participants’ experience during the first lockdown. The study obtained approval from the Ethics Committee of University College Dublin.

### 2.2. Participants

Participants were 579 parents with at least one minor child (ranging from 0 to 18 years of age), 88% of which were females. The parents’ age range was 25–63 years (M = 45.10, SD = 6.32). The majority of participants, 68.1% (n = 439), were residents of Italy, 31.9% (n = 140) were residents in the Republic of Ireland, and one was in Northern Ireland; 87.8% stated they were living with their partner; the majority (52.5%) reported having two children, 26.5% stated they had one child, 15.9% three children, 3.5% four children, and the minority had five (0.7%) or more than five (0.9%) children. The sample’s education level could be classified as medium-high since 60.1% of participants had obtained a Bachelor’s or higher degree. Additionally, the mean age of the Irish sample was lower (M = 43.4 years) compared with the Italian sample (M = 48.1). An independent t-test showed that the difference between the two countries was significant (*t* = 2.08, df = 577, *p* = 0.04, two-tailed, *d*= 26.46). Furthermore, the level of education resulted in being significantly different across the two samples (*t* = −4.44, df = 472, *p* = < 0.001, two-tailed, *d*= 1.04), since the Irish sample showed a higher level of education (M = 4.26) compared with the Italian sample (M = 3.87).

### 2.3. Measures

*Parents’ well-being.* Parents’ well-being was assessed using the Warwick-Edinburgh Mental Well-being Scale (WEMWBS) [60,61], a 12-items 5-point Likert-type from ‘never’ to ‘always’ asking about individual feelings (e.g., ‘I’ve been feeling optimistic about the future’). Cronbach’s alpha coefficient for this scale was 0.88.

*Technology interference*. Family conflict over the use of digital media was measured using an adaptation of the 4-items interpersonal conflict (IC) sub-scale from the Generic Scale of Phubbing (GSP) [62]. The IC is a 5-point rating scale ranging from ‘never’ to ‘always’ asking about the frequency of conflict between the parental couple and between parent and children over the use of technologies, e.g., ‘I tell my partner that s/he interact with her/his smartphones (or another device connected to Internet, e.g., tablet) too much’ and ‘I tell my child that s/he interact with her/his smartphones (or another device connected to Internet, e.g., tablet) too much’. Cronbach’s alpha coefficient for parent–partner IC was 0.90, whereas for parent–child, IC was 0.87.

*Family resilience*. The Walsh Family Resilience Questionnaire (WFRQ) [55,63] was used as a scale to measure family resilience. The 31-item questionnaire on a Likert-type scale from 1 = ‘very little’ to 5 = ‘very much’ is a well-validated instrument to assess the three-factors structure of the family resilience framework [64], namely belief systems, communication processes, and organizational resources. Some items were adapted to refer specifically to the COVID-19 pandemic situation (e.g., ‘we trust in the possibility of overcoming our difficulties brought by this pandemic’). The scale showed very good reliability (Cronbach’s alpha= 0.93).

*Parents’ domestic workload*. Parents’ domestic workload was measured by asking participants to estimate the number of hours dedicated to (a) domestic activities, such as cooking, tiding up, grocery shopping, (b) playing with children, and (c) helping children with homework/study in a typical day during the lockdown.

*Digital media use*. The extent to which participants used technologies (devices that support the Internet) during the lockdown was measured by four items asking them the frequency of use of smartphone, computer, tablet, and TV for job or leisure activities on a range from 1 = never to 5 = very often.

*Children’s age and number*. Participants were asked to indicate the number of children present in the household and their age.

*Country of residence*. This was assessed with a single item asking participants to report whether they lived in either the Republic of Ireland or Italy.

### 2.4. Data Analysis Overview

A composite variable was created to measure the frequency of digital media use during the pandemic (smartphone, computer, laptop, TV). Principal Component Analysis was applied using the regression scores approach. This method allows the composite variable to reflect the latent dimension of device usage by encompassing the structure of the initial items that assessed frequency of use of each device [65]. A one component solution best summarized device usage in the data based on an eigenvalue of 1.63 and accounted for 41% of the variance (component loadings ranged from 0.52 to 0.76) with the second component producing an eigenvalue lower than 1.0 (0.95).

Descriptive statistics were calculated for the study variables. A hierarchical regression analysis was conducted to evaluate the significance and strength of the predictive value of the following variables on parents’ well-being: (a) family conflict over the use of digital media (*technoference*), (b) frequency of use of digital media, (c) family resilience, (d) children’s age, and € number of children present in the household. Model predictors were entered in three steps. In Step 1, age, country of residence (Italy or Republic of Ireland), and number of children in the family were entered as predictors, and in Step 2, family resilience was entered in the model. In Step 3, variables related to use of digital media were entered as predictors, namely device usage, conflict between parents over the use of technology, parent–child conflict over the use of technology, and parents’ workload. In the second and third steps, a significant change of R^2^ value indicates a significant contribution of the group of predictors to the total amount of variance above and beyond the predictor entered in Step 1. Confidence Intervals (CIs) and alpha were set to 95% and 0.05, respectively.

## 3. Results

### 3.1. Descriptive Statistics

As shown in Table 1, participants had, on average, two children, and children’s mean age was 11.63 years. Parents reported a medium to high score on family resilience of 3.60. As for the frequency of devices used during home confinement, smartphone was the most frequently used device (M = 4.38), followed by computer (M = 4.05) and TV (M = 3.52,). Tablets were used rarely (M = 2.68). Participants reported low levels of couple conflict over technology use (M = 1.78) and medium levels of parent–child conflict over technology use (M = 2.59). Regarding domestic workload, parents reported spending, on average, 5.20 h doing domestic activities, 3.83 h playing with children, and 3.37 h helping children with homework daily. Lastly, parents reported moderate levels of well-being (M = 3.42).

Additionally, we statistically tested for differences across all variables between Italian and Irish parents (please see Table 1 for additional details). Notably, Irish parents reported higher levels of family resilience (*p* < 0.001, η^2^ = 0.06), lower levels of parents’ well-being (*p* = 0.002, η^2^ = 0.02), and higher levels of parental couple conflict over technology (*p* < 0.001, η^2^ = 0.03) than Italian parents. However, the small effect sizes indicate that the differences in parental conflict and parents’ well-being between the two samples were not substantial. Similarly, significant but small differences emerged between the two samples regarding parents’ domestic workload, smartphone use, and number of children. Italian parents reported higher levels of involvement in domestic activities (*p* < 0.001, η^2^ = 0.02) and helping children with homework (*p* < 0.001, η^2^ = 0.04) compared with Irish parents. Irish parents reported a higher frequency of smartphone use during lockdown (*p* < 0.01, η^2^ = 0.02) than Italian parents. Irish parents reported a higher number of children (*p* < 0.05, η^2^ = 0.01) compared with Italian parents. Lastly, a larger effect size was observed for the variable ‘age of children’: Irish parents had younger children than Italian parents (*p* < 0.001, η^2^ = 0.14) (See Table 1 below for details).

### 3.2. Hierarchical Regression

Table 2 shows the results of the hierarchical regression model. In Step 1 of the hierarchical regression model, age, country of residence (Italy or Republic Ireland), and number of children were entered as predictors. The results showed that together these variables produced a significant contribution in predicting parents’ well-being F(3, 564) = 5.38, *p* = 0.001. Despite a significant *p*-value, the variance explained in scores of the outcome variable was substantially small (2%). A higher age of children significantly predicted better parents’ well-being (*p* = 0.04) and living in the Republic of Ireland (*p* = 0.03), while the number of children was not a significant individual predictor (*p* = 0.31).

In relation to parents’ well-being, family resilience was entered as predictor variable in the second step of the regression model. The results showed that family resilience produced a considerable contribution to the model F(4, 564) = 86.4, with *p* < 0.001 explaining 38% of the variance. Entering family resilience as a predictor added to the explained variance of parents’ well-being above and beyond the contribution of age, country of residence, and number of children R^2^ change = 0.35, F(1, 560) = 320.4, *p* < 0.001. Family resilience was a strong positive predictor of parents’ well-being (β = 0.62, *p* < 0.001), and country of residence and age of children remained significant predictors in this step (see Table 2).

Variables related to the use of digital media (digital media use, conflict between parents over digital media use, and conflict between parent and children over digital media use) and workload (in hours) were added in Step 3 of the hierarchical regression model. Together these variables produced a significant contribution to parents’ well-being. As shown in Table 2, these variables added a small contribution to the explained variance of parents’ well-being above and beyond the contribution of family resilience R^2^ change = 0.02, F(4, 556) = 4.16, *p* = 0.002, increasing the explained variance to 39%. Of the four predictors inserted at the third step, only conflict between parents was a significant unique predictor (*p* < 0.001), while digital media use (*p* = 0.99), parent–child conflict (*p* = 0.25), and workload (*p* = 0.78) were not.

## 4. Discussion

This study aimed to provide a better understanding of parent–child relationships and parents’ well-being under the unprecedented circumstances induced by the COVID-19 pandemic, such as home confinement. Research has pointed out that the impact of parents’ stress on parents’ well-being can spill over parent–child relationships and ultimately negatively affect children’s adjustment and well-being, e.g., [1,2,5,6,7,8]. Our findings showed that a key protective factor for parents’ well-being during the COVID-19 restrictions and home confinement was family resilience. The more the parents perceived that they could work together as a family to overcome the challenges of the pandemic in terms of hope and positive belief, positive interpersonal communication, and mobilization of internal and external resources, the less parents’ well-being was affected. In other words, family resilience worked as a ‘buffer’ that prevented parents from being stressed or overwhelmed by the negative circumstances of home confinement [27,56]. When families are resilient, parents’ well-being is safeguarded, and this ultimately can protect parent–child relationships and children’s well-being.

Another interesting result that emerged from our study was related to the impact of the use of digital media on parents’ well-being during lockdown. The frequency of use of digital media did not have an impact on parents’ well-being; however, the distraction and interruptions of face-to-face communication caused by the use of digital media triggered conflicts between the partners (*technoference*), which in turn impacted parents’ well-being. Notably, parent–child conflict over the use of digital devices had no impact on parents’ well-being. Evidence shows that the interference of technologies in face-to-face communication negatively affects the quality of family relationships [52,53,54], however, our results suggest that parents are more negatively affected by conflict with their partner rather than with their children. This result can be linked to the contextual circumstances of the lockdown, namely, parents could only rely on one another and work as a team to cope with the multiple challenges of the pandemic. In this sense, one parent’s continuous distraction due to the use of digital media may have prevented the parents from being available for one another, both as parents and as partners. Therefore, it is arguable that digital devices have undermined the possibility for parents to collaborate and to spend time together as a couple, thereby negatively affecting parents’ well-being, as our data documented.

Cross-country differences between Ireland and Italy were also considered in our analyses. No substantial differences were found between the two samples regarding *technoference*; instead, the two samples significantly differed on the perception of family resilience. Irish parents reported higher levels of perceived resilience compared with Italian parents: Resilience is a key contributor to parents’ well-being; therefore, it is arguable that Irish parents have better coped with COVID-19 lockdown. This result could be interpreted considering the characteristics of the sample: Irish parents have, on average, a higher level of education, and that might have facilitated them in mobilizing resources to face COVID-19 challenges. However, contextual factors might also have favored Irish parents since Italians faced stricter restrictions, such as the limitation of movement and this might have exacerbated parents’ stress and affected family resilience. Furthermore, although the mean age of children was different between two samples, with Italian parents reporting higher mean scores, the amount of variance explained in the total model of hierarchical regression (Step 1, results Section 3.2) indicates that this was not a considerable unique predictor of parents’ well-being.

Lastly, in contrast with previous studies [38], the number of children present in the household, as well as the domestic workload, were not associated with parents’ well-being [23,44]. Instead, in line with existing evidence [22,37], having young children negatively impacted parents’ well-being. Differently from parents with adolescents or older children, parents with young children had to be involved in continuous child-minding and daily care routines (food preparation, hygiene, homework, and playful activities), while contemporarily being involved in other tasks, such as housekeeping and remote work activities. The management of different and multiple tasks at a time, for the lack of support provided by educational agencies and extended families, might have increased parents’ level of stress that ultimately impacted their well-being and the relationship with their children.

## 5. Limitations and Conclusions

This study has some limitations. Firstly, a perceived measure of parents’ stress was not included in our analysis. This can be an additional measure to include in future studies that want to consider the long-term impact of the pandemic on parent–child relationships. Secondly, we could not rely on measures referred to families’ contextual situation before the pandemic e.g., the use of digital media and the conflict over the use of devices. This could have provided more details about the possible amplification of dysfunctional family processes and the impact on parents’ and children’s wellbeing. Our study could only provide a snapshot of processes that happened during a situation of acute stress. Future studies should consider longitudinal research designs for monitoring potential changes in families’ coping mechanisms, as well as their impact on family members’ well-being and their relationships. Thirdly, future studies should further investigate the single contribution of family resilience dimensions to parents’ well-being. Finally, because more participants were recruited from Italy than from the Republic of Ireland, this may have created a sample imbalance. Thus, our findings should be interpreted with caution, especially when considering cross-country differences.

The COVID-19 pandemic has brought several challenges for families, including mandatory confinement at home for several weeks. As indicated by our findings, parents had to cope with contextual factors that may have affected their well-being, as well as their couple and parent–child relationships. At the time of writing this article (October 2022), it seems that we have entered a post-pandemic era; however, coronavirus has not disappeared, and parents and children are now coping with the long-term impact of this catastrophic event. Therefore, practitioners working with families should be aware of the conflictual dynamics that parents and children have faced during the pandemic. Digital media became an issue of confrontation between the parental couple, thereby negatively affecting parents’ well-being. The consideration of the quality of a couple’s relationship is therefore key when devising clinical interventions. Additionally, our results showed that family resilience framework continues to be an important conceptual framework that can guide individual and family interventions during the post-pandemic era.

## Figures and Tables

**Table 1 ijerph-19-15687-t001:** Descriptive statistics and ANOVA results for the variables included in the study.

Variables	Sample	N	M (SD)	df	F	*p*	η^2^
Age of children	Total	568	11.63 (5.40)	1	89.70	<0.001	0.14
	Irish	203	8.96 (4.96)				
	Italian	365	13.12 (5.05)				
Number of children	Total	573	2.02 (0.88)	1	3.99	0.05	0.01
	Irish	206	2.12 (1.00)				
	Italian	367	1.96 (0.79)				
Family Resilience	Total	579	3.60 (0.55)	1	34.43	<0.001	0.06
	Irish	206	3.77 (0.61)				
	Italian	373	3.50 (0.48)				
Device usage:							
Smartphone	Total	579	4.38 (0.75)	1	8.55	<0.01	0.02
	Irish	206	4.50 (0.70)				
	Italian	373	4.32 (0.77)				
Computer	Total	579	4.05 (1.07)	1	2.73	0.10	0.01
	Irish	206	4.15 (1.14)				
	Italian	373	3.99 (1.03)				
TV	Total	579	3.52 (1.04)	1	1.34	0.25	0.00
	Irish	206	3.59 (1.26)				
	Italian	373	3.48 (0.90)				
Tablet	Total	579	2.68 (1.44)	1	3.37	0.07	0.01
	Irish	206	2.83 (1.47)				
	Italian	373	2.60 (1.43)				
Parental couple conflict over digital media	Total	579	1.78 (1.75)	1	18.53	<0.001	0.03
	Irish	206	1.96 (0.82)				
	Italian	373	1.68 (0.69)				
Parent–child conflict over digital media	Total	579	2.59 (0.92)	1	2.54	0.08	0.01
	Irish	206	2.68 (0.97)				
	Italian	373	2.54 (0.88)				
Parents’ domestic workload:							
Domestic activities	Total	579	5.20 (4.74)	1	13.13	<0.001	0.02
	Irish	206	4.25 (2.91)				
	Italian	373	5.73 (5.43)				
Playing with children	Total	579	3.83 (4.52)	1	3.66	0.06	0.01
	Irish	206	3.35 (3.37)				
	Italian	373	4.10 (5.03)				
Helping children with homework	Total	579	3.37 (4.77)	1	24.19	<0.001	0.04
	Irish	206	2.08 (2.86)				
	Italian	373	4.08 (5.43)				
Parents’ well-being	Total	579	3.42 (0.56)	1	9.98	0.002	0.02
	Irish	206	3.32 (0.61)				
	Italian	373	3.48 (0.52)				

**Table 2 ijerph-19-15687-t002:** Hierarchical regression analyses for predictors of family well-being.

Step	Predictor	B	*SE*	β	R^2^ (Adjusted)	R^2^ Change
1					0.02 ** (*p* = 0.001)	
	Age of children	0.01	0.004	0.1 ** (*p* = 0.04)		
	Country of residence	0.12	0.05	0.09 ** (*p* = 0.03)		
	Number of children	0.03	0.03	0.03		
2					0.38 *	0.36 *
	Age of children	0.009	0.004	0.09 ** (*p* = 0.01)		
	Country of residence	0.29	0.04	0.24 *		
	Number of children	−0.01	0.03	−0.02 (ns)		
	Family resilience	0.62	0.04	0.62 *		
3					0.39 *	0.02 ** (*p* = 0.002)
	Age of children	0.008	0.004	0.08 ** (*p* = 0.02)		
	Country of residence	0.26	0.04	0.22 *		
	Number of children	−0.02	0.03	−0.02 (ns)		
	Family resilience	0.62	0.04	0.60 *		
	Digital media use	−0.001	0.02	−0.001 (ns)		
	Parental couple conflict	−0.10	0.03	0.14 *		
	Parent–child conflict	0.02	0.02	0.04 (ns)		
	Parents’ domestic workload (h)	0.001	0.002	0.009 (ns)		

Note: * *p* < 0.001; ** *p* < 0.05; ns, not significant; country of residence, Republic of Ireland or Italy.

## Data Availability

No data are made available for open access.

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
