# Peer review of "Parent–Child Relationships, Digital Media Use and Parents’ Well-Being during COVID-19 Home Confinement: The Role of Family Resilience"

_ijerph, 2022, doi:10.3390/ijerph192315687_

Round 1

Reviewer 1 Report

Interesting article overall, but requires major edits with regards to re-organizing the sections, and reviewing for grammar and conciseness. 

Introduction:

Lines 31-34 are wordy and unclear, please make more concise: Similarly, adolescence can be a period of stress and conflicts for parents and children who are called to renegotiate families’ rules and roles, intimacy, and interpersonal distances, and acknowledge children’s emerging competencies.

Section 1.1

Line 76 - define "vulnerable families" 

Lines 98-102, wordy and long-winded, please make more concise: This phenomenon has offered parents and children opportunities, such as exploring and using together online platforms, using online services for healthcare, work, and education, which were not available pre-COVID; however, increased screen time had also implied increased exposure to risks and harm [45, 46].

Methods

It's unclear why the two participating countries (i.e., Italy and Ireland) were selected for this study. The authors compare the lockdown measures in both places; however, these measures were in place worldwide. Ireland seems to be the host country, but why was Italy selected? And, why Northern Italy?

The authors have blended results and methods together in the section under 'materials and methods'. For instance, the 'participants' section should be listed under Results whereas Data Analysis Overview should be under methods. Please rework the entire materials and methods to divide into clear methods and results sections.

Discussion

Line 292: Please rephrase 'an innovative result' phrasing - this wording does not make sense. 

Line 293: Remove 'per se'.

Lines 294-295 should read as follows: however, the distraction and interruptions of face-to-face communication caused by the use of digital media (technoference) triggered conflicts between the partners, which in turn impacted on parents’ well-being. (Remove on)

Line 297: parent-child conflict for the use of digital devices - rewording to: parent-child conflict over the use of digital devices is more clear.

Same feedback for line 303: In this sense, one parent’s continuous distraction for the use of digital media may have prevented the parent from being available for the other. 'For' does not make sense here, 'due to' is better or something similar. 

Line 307: "the number of children present in the household, as well as the domestic workload, were not associated" - need commas. This is a common trend throughout the paper, grammar needs to be addressed.

Line 309: having young children negatively impacted on parents’ well-being - 'on' not needed.

Line 316: This study has some limitations: Firstly, - firstly should not be capitalized, and the colon is improperly used here because you have not written an on-going list. Please revise. 

Line 319: e.g., goes in brackets

Line 321: The authors state they collected data during acute crisis - though it may seem obvious, please clarify why this is a limitation and how it may skew data. It is not enough to just list it.

Please review this: However, during the pandemic, Italy and Ireland faced similar restriction measures, and research evidence studies highlighted parents’ responses during the pandemic did not significantly differ across culture [17]. 

First, the authors need to use commas. Second, the highlighted portion does not make sense. Lastly, the paper by Toran et al. [17] notes that "China and Turkey share significant similarities in parenting cultures". This indicates that the two populations were already similar in their parenting styles/responses, so major differences across the two cultures during the pandemic would have been surprising. And, the authors cannot generalize that all parental responses during the pandemic did not differ across cultures. Either specify that this study was conducted in Turkey and China, or rephrase the whole sentence. 

Author Response

Introduction:

Lines 31-34 are wordy and unclear, please make more concise: Similarly, adolescence can be a period of stress and conflicts for parents and children who are called to renegotiate families’ rules and roles, intimacy, and interpersonal distances, and acknowledge children’s emerging competencies.

Response: we rewrote this sentence.

Section 1.1

Line 76 - define "vulnerable families" 

Response: This definition was rephrased

Lines 98-102, wordy and long-winded, please make more concise: This phenomenon has offered parents and children opportunities, such as exploring and using together online platforms, using online services for healthcare, work, and education, which were not available pre-COVID; however, increased screen time had also implied increased exposure to risks and harm [45, 46].

Response: we rewrote this sentence

Methods

‘It's unclear why the two participating countries (i.e., Italy and Ireland) were selected for this study. The authors compare the lockdown measures in both places; however, these measures were in place worldwide. Ireland seems to be the host country, but why was Italy selected? And, why Northern Italy?’

Response: This study was part of a larger project that involved two countries. Details were added in a footnote.

The authors have blended results and methods together in the section under 'materials and methods'. For instance, the 'participants' section should be listed under Results whereas Data Analysis Overview should be under methods. Please rework the entire materials and methods to divide into clear methods and results sections.

Response: Thank you for addressing this aspect. We have now added the results section and amended the subheadings accordingly.

Discussion

Line 292: Please rephrase ‘an innovative result’ phrasing – this wording does not make sense. 

Line 293: Remove ‘per se’.

Lines 294-295 should read as follows: however, the distraction and interruptions of face-to-face communication caused by the use of digital media (technoference) triggered conflicts between the partners, which in turn impacted on parents’ well-being. (Remove on)

Line 297: parent-child conflict for the use of digital devices – rewording to: parent-child conflict over the use of digital devices is more clear.

Same feedback for line 303: In this sense, one parent’s continuous distraction for the use of digital media may have prevented the parent from being available for the other. ‘For’ does not make sense here, ‘due to’ is better or something similar. 

Line 307: “the number of children present in the household, as well as the domestic workload, were not associated” – need commas. This is a common trend throughout the paper, grammar needs to be addressed.

Line 309: having young children negatively impacted on parents’ well-being – ‘on’ not needed.

Line 316: This study has some limitations: Firstly, - firstly should not be capitalized, and the colon is improperly used here because you have not written an on-going list. Please revise. 

Line 319: e.g., goes in brackets

Response: All these aspects have been amended.

Line 321: The authors state they collected data during acute crisis – though it may seem obvious, please clarify why this is a limitation and how it may skew data. It is not enough to just list it.

Response: we specified that the limitation is in the lack of measures about family functioning dimensions before the pandemic.

Please review this: However, during the pandemic, Italy and Ireland faced similar restriction measures, and research evidence studies highlighted parents’ responses during the pandemic did not significantly differ across culture [17]. 

Response: This paragraph was changed substantially.

First, the authors need to use commas. Second, the highlighted portion does not make sense. Lastly, the paper by Toran et al. [17] notes that “China and Turkey share significant similarities in parenting cultures”. This indicates that the two populations were already similar in their parenting styles/responses, so major differences across the two cultures during the pandemic would have been surprising. And, the authors cannot generalize that all parental responses during the pandemic did not differ across cultures. Either specify that this study was conducted in Turkey and China, or rephrase the whole sentence. 

Response: Further analysis considering also cross-country dimensions allowed us to clarify this aspect. Thank you very much for addressing this issue.

Reviewer 2 Report

This paper examined what variables explain parents’ wellbeing during lockdown in Italy and Ireland using questionnaire data from parents. This is a timely topic. Data analyses need to be revised to consider the cultural contexts (e.g., in terms of measurement invariance). In addition, the

Literature Review

- p.2, ol.58-59: Due to the scope of the present study, I would prefer to focus on parents’ wellbing as outcome in this sentence (and delete hitns to adjustment and parent-child relationships as other possible outcomes).

- I am missing the rationale for why the authors focus on parents’ wellbeing. You need to further address this issue in the literature review. It would be nice if this could also be justified conceptually/theory-based.

Method

- Participants: Information about the participants needs to be provided for Italian and Irish each. Furthermore, it is necessary to test that both do not differ in their sociodemographic background. On the contrary, it is not possible to use the data of Irenes and Italians as one sample in the analyses.

- Did you check if there were differences between fathers and mothers (e.g., scale means)?

- Was the questionnaire for Italians in Italian and for Irish in English? If yes, how did you check for comparability of the questionnaires?

- Measures - scale to assess digital media use: The response format is problematic because "a lot" and "a little" do not mean the same thing to all people.

- 2.4: I am not sure if it is the right decission to use digital media use as one scale. From similiar questinnaire items elsewhere I know that it usually not the case that different media can be used as a reliable scale. In addition, the authors do not report Cronbach’s Alpha for their digital media use scale. Can you please check reliability of this scale? And if the scale is not reliable, I recommend to insert the single items in the regression. This may change the results, too. In Table 1 we see that almost everyone is using smartphones, but tablets and TVs are less used.

- With chapter 2.5 the results section starts. Please add a heading „Results“.

Results

- Table 1: Again, we need to see the data for Italy and Ireland each, too. Otherwise, it is not possible to check if it is similar between countries.

- Chapter 2.5: It is not necessary to repeat all M and SD in the text that are already written in the Table, rather the authors can refer to Table 1.

Table 1: Beta cannot be greater than 1. The same counts vor R². Thus, please delete the zero before the decimal separator.

- Table 1: In step 2, there is a beta of .55. To which predictor does this beta coefficient belong?

- I think it would be necessary to include the country at least as control variable into the regression analyses - even if it should turn out that the mean values of the scales and sociodemographic variables do not differ between Italians and Irenes.

Discussion

- I think for parents changed a lot and that the questionnaire aimed on explaining parents’ wellbeing including a couple of family-/child-related variables but less variables related to the partner. This may be a point the authors can discuss also with a view on further reserach.

- Regarding the limitation of neclecting the cross-cultural dimension I want to see here some changes after re-analyzing the data considering the country in the analyses.

- p. 8, l. 339: This statement is not consistent with the results, which showed no effect of couple conflicts on parents' wellbeing.

- The discussion comes up a bit short for me in terms of what we learn from it for theory development as well: To what extent do the results go hand in hand with theories of wellbeing and (family) resilience?

- And one could also conclude with why parents wellbeing is so important - maybe for the parents themselves, but also for the children and for the relationships in the families?

- Data Availability Statement: To advance open science, I strongly recommend to provide the dataset inclusive analysis script Open Access via the journal or an repository (e.g., osf.io).

- Author Contributions: There is some text that needs to be deleted.

- Institutional Review Board Statement: There is some text that needs to be deleted.

Author Response

This paper examined what variables explain parents’ wellbeing during lockdown in Italy and Ireland using questionnaire data from parents. This is a timely topic. Data analyses need to be revised to consider the cultural contexts (e.g., in terms of measurement invariance). In addition, the

Response: Thank you very much for this suggestion. We run further analysis considering cross-country dimensions. This has allowed us to elaborate further on our results, thereby improving the quality of the manuscript.

 Literature Review

- p.2, ol.58-59: Due to the scope of the present study, I would prefer to focus on parents’ wellbing as outcome in this sentence (and delete hitns to adjustment and parent-child relationships as other possible outcomes).

Response: The sentence was amended as indicated.

- I am missing the rationale for why the authors focus on parents’ wellbeing. You need to further address this issue in the literature review. It would be nice if this could also be justified conceptually/theory-based.

Response: The literature review was integrated with more references related to parental stress, spill over effect, and coping mechanisms. We hope the Introduction section is clearer and better linked to the rationale of this study.

Method/Results

- Participants: Information about the participants needs to be provided for Italian and Irish each. Furthermore, it is necessary to test that both do not differ in their sociodemographic background. On the contrary, it is not possible to use the data of Irenes and Italians as one sample in the analyses.

Response: Thank you for this comment. We provided additional information on the sociodemographic background for both samples, Irish and Italians. We conducted further analysis controlling for country of residence which allows to use the sample as one in hierarchical regression models.

- Did you check if there were differences between fathers and mothers (e.g., scale means)?

Response: Thank you for bringing that aspect to our attention. However, it is not among the scopes of this article to consider mother an fathers’ differences, since our interest was on parenting and couple dynamics.

- Was the questionnaire for Italians in Italian and for Irish in English? If yes, how did you check for comparability of the questionnaires?

Response: The questionnaire was developed first in Italian and then translated in English. Three authors are native Italian speakers but have worked and lived in English speaking countries for more than 5 years. A first level translation was carried out by the authors who then a native English speaker colleague was involved in refining the translation in English.

- Measures - scale to assess digital media use: The response format is problematic because "a lot" and "a little" do not mean the same thing to all people.

Response: As reported in the manuscript, page - line 209, the response format of the digital media use scale is 1=never to 5=very often.

- 2.4: I am not sure if it is the right decission to use digital media use as one scale. From similiar questinnaire items elsewhere I know that it usually not the case that different media can be used as a reliable scale. In addition, the authors do not report Cronbach’s Alpha for their digital media use scale. Can you please check reliability of this scale? And if the scale is not reliable, I recommend to insert the single items in the regression. This may change the results, too. In Table 1 we see that almost everyone is using smartphones, but tablets and TVs are less used.

Response: Thank you for raising this. However, we have not used the digital media as one scale but as a composite variable (please see section 2.4 for a detailed presentation on how this was statistically created), which is an approach superior addressing reliability/validity issues resulting from putting together single items as one scale. Cronbach’s alpha cannot be calculated for a composite variable.

- With chapter 2.5 the results section starts. Please add a heading „Results“.

Response: Thank you very much for noting that. We inserted that section.

Chapter 2.5: It is not necessary to repeat all M and SD in the text that are already written in the Table, rather the authors can refer to Table 1.

Response: This has been addressed in the revised manuscript.

Table 1: Beta cannot be greater than 1. The same counts vor R². Thus, please delete the zero before the decimal separator.

Response: We believe that the Reviewer refers to Table 2, where the regression results are presented. No Beta coefficients are greater than one as shown in Table 2. But even if there were > 1, these are unstandardized coefficients, so it is absolutely normal to have coefficients > 1.  No R² is > 1 as shown in Table 2. Please note that zero before the separator adheres to journal’s reporting style, for instance 0.55 instead of .55.

- Table 1: In step 2, there is a beta of .55. To which predictor does this beta coefficient belong?

Response: Thank you for noticing this. This has been deleted from Table 2 in the revised manuscript.

I think it would be necessary to include the country at least as control variable into the regression analyses - even if it should turn out that the mean values of the scales and sociodemographic variables do not differ between Italians and Irenes.

Response: This has been addressed, we controlled for country in the regression models. Please see reporting of results in the revised manuscript and relevant section in the discussion.

Discussion

- I think for parents changed a lot and that the questionnaire aimed on explaining parents’ wellbeing including a couple of family-/child-related variables but less variables related to the partner. This may be a point the authors can discuss also with a view on further reserach.

Response: this was added to the limitations and conclusion session.

- Regarding the limitation of neclecting the cross-cultural dimension I want to see here some changes after re-analyzing the data considering the country in the analyses.

Response: thank you for this suggestion. We run the analysis including the country as an additional variable.

- p. 8, l. 339: This statement is not consistent with the results, which showed no effect of couple conflicts on parents' wellbeing.

Response: The statement was rewritten.

- The discussion comes up a bit short for me in terms of what we learn from it for theory development as well: To what extent do the results go hand in hand with theories of wellbeing and (family) resilience?

Response: in the discussion session, we specified that family resilience is a protective factor for parents’ well-being. Also, we elaborated further on theories on family stress and spillover effect.

- And one could also conclude with why parents wellbeing is so important - maybe for the parents themselves, but also for the children and for the relationships in the families? 

Response: Thank you for noting this. We have rewritten this section and made it more coherent with the additional references we inserted.

- Data Availability Statement: To advance open science, I strongly recommend to provide the dataset inclusive analysis script Open Access via the journal or an repository (e.g., osf.io).

Response: We haven't received consent of participants to make their data available through a public open access repository, therefore we cannot allow the access to the dataset.

- Author Contributions: There is some text that needs to be deleted.

Response: Thank you for noting this. We have now deleted it.

- Institutional Review Board Statement: There is some text that needs to be deleted.

Response: Thank you for noting this. We have now deleted it.

Round 2

Reviewer 1 Report

The revisions have made the manuscript more clear and concise. There are still a few typos throughout - reading through once more would ensure you correct them prior to publication. 

Author Response

Thank you very much for the feedback.

The article was proofread and typos corrected.

Reviewer 2 Report

- Method: Thank you for answering my question on the language of the questionnaire. Can you please included this information into the manuscript?

- Table 2: Shouldn't it be "parent's well-being" instead of "family well-being"?

- Table 2: There are significant effects of age of cildren and country of residence on parent's well-being. I guess there may be an interaction effect. Did you check this? I think that needs to be reported.

- Discussion: You should discuss the response format 1=never to 5=very often as not everyone understands the same as "very often" - how often is "very often"? Each day? Every second day? Multiple times a day?....

- Discussion: You should discuss the importance of multilevel analyses for comparing the different countries. It seems, that your sample was not balanced and there might be substantial variance on country level. 

- For future studies, I strongly recommend to get consent that you can share the anonymze data!

Author Response

Thank you very much for the additional comments.

Details about the translation of the questionnaire were added as a footnote.

Table 2: There are significant effects of age of cildren and country of residence on parent's well-being. I guess there may be an interaction effect. Did you check this? I think that needs to be reported.

Response: Lines 305-36 of the manuscript: As noted the predictors in Step 1 explained only 2% of the variance in scores in wellbeing, which cannot (and should not) allow any further inferences but those we already presented in the results. However, we added a relevant statement in the discussion in round 2 of revisions, please see lines 375-379.

Discussion: You should discuss the response format 1=never to 5=very often as not everyone understands the same as "very often" - how often is "very often"? Each day? Every second day? Multiple times a day?....

Response: This 5-point Likert scales with these response options are commonly and widely used in research because they allow to capture a range of responses on the frequency of use. More specifically the range was: never, rarely, sometimes, often, very often. This scale was used in previous studies (Everri, 2017; Everri, Messena, Mancini, 2019).

Discussion: You should discuss the importance of multilevel analyses for comparing the different countries. It seems, that your sample was not balanced and there might be substantial variance on country level. 

Response: We have added this as a potential limitation of the study, please see lines 415-417.